# Combined Tumor-Based *BRCA1/2* and *TP53* Mutation Testing in Ovarian Cancer

**DOI:** 10.3390/ijms241411570

**Published:** 2023-07-18

**Authors:** Edith Borcoman, Elizabeth Santana dos Santos, Catherine Genestie, Patricia Pautier, Ludovic Lacroix, Sandrine M. Caputo, Odile Cabaret, Marine Guillaud-Bataille, Judith Michels, Aurelie Auguste, Alexandra Leary, Etienne Rouleau

**Affiliations:** 1Department of Medical Oncology, Institut Curie, 75005 Paris, France; 2Department of Drug Development and Innovation (D3i), Institut Curie, 75005 Paris, France; 3Department of Medical Oncology, A.C.Camargo Cancer Center, São Paulo 01509-010, Brazil; 4Department of Medical Biology and Pathology, Gustave Roussy, 94805 Villejuif, France; catherine.genestie@gustaveroussy.fr (C.G.); ludovic.lacroix@gustaveroussy.fr (L.L.); odile.cabaret@gustaveroussy.fr (O.C.); marine.guillaud-bataille@gustaveroussy.fr (M.G.-B.); 5INSERM U981, Translational Research Laboratory, University Paris-Saclay, 94805 Villejuif, France; aurelie.auguste@gustaveroussy.fr (A.A.); alexandra.leary@gustaveroussy.fr (A.L.); 6Gynecology Unit, Gustave Roussy, 94805 Villejuif, France; patricia.pautier@gustaveroussy.fr (P.P.); judith.michels@gustaveroussy.fr (J.M.); 7Groupe d’Investigateurs Nationaux pour l’Etude des Cancers Ovariens (GINECO), 94805 Villejuif, France; 8Department of Genetics, Institut Curie, PSL Research University, 75005 Paris, France; sandrine.caputo@curie.fr

**Keywords:** *BRCA1/2* tumoral testing, *TP53m*, ovarian cancer, high-grade serous ovarian cancers, loss of heterozygosity, allelic frequency

## Abstract

Somatic/germline *BRCA1/2* mutations (m)/(likely) pathogenic variants (PV) (*s*/*gBRCAm*) remain the best predictive biomarker for PARP inhibitor efficacy. As >95% of high-grade serous ovarian cancers (HGSOC) have a somatic *TP53m*, combined tumor-based *BRCA1/2* (*tBRCA*) and *TP53* mutation testing (*tBRCA/TP53m*) may improve the quality of results in somatic *BRCAm* identification and interpretation of the ‘second hit’ event, i.e., loss of heterozygosity (LOH). A total of 237 patients with HGSOC underwent *tBRCA/TP53m* testing. The ratio of allelic fractions (AFs) for *tBRCA/TP53m* was calculated to estimate the proportion of cells carrying *BRCAm* and to infer LOH. Among the 142/237 *gBRCA* results, 16.2% demonstrated a pathogenic/deleterious variant (DEL) *gBRCA1/2m*. Among the 195 contributive tumor samples, 43 DEL of *tBRCAm* (22.1%) were identified (23 *gBRCAm* and 20 *sBRCAm)* with LOH identified in 37/41 conclusive samples. The median AF of *TP53m* was 0.52 (0.01–0.93), confirming huge variability in tumor cellularity. Initially, three samples were considered as wild type with <10% cellularity. However, additional testing detected a very low AF (<0.05) in both *BRCA1/2m* and *TP53m*, thus reidentifying them as *sBRCA1/2m*. Combined *tBRCA/TP53m* testing is rapid, sensitive, and identifies somatic and germline *BRCA1/2m.* AF *TP53m* is essential for interpreting *sBRCA1/2m* in low-cellularity samples and provides indirect evidence for LOH as the ‘second hit’ of *BRCA1/2*-related tumorigenesis.

## 1. Introduction

Germline mutations (m)/(likely) pathogenic variants in *BRCA1* or *BRCA2* (*BRCA1/2m*) genes are well-established causes of breast and ovarian cancer genetic predisposition, leading to deficiency in the homologous recombination repair pathway (HRD), where *BRCA1* and *BRCA2* are involved in the efficient reparation of DNA double-strand breaks [1]. It is currently established that hereditary predispositions are present in approximately 25% of ovarian cancer cases [2].

Based on the concept of synthetic lethality, by which cell death results from the inactivation of two genes when inactivation of either gene alone is nonlethal [3], poly (ADP-ribose) polymerase (PARP) inhibitors have been developed to inhibit the reparation of DNA single-strand breaks, showing improvement of survival in high-grade serous ovarian cancers (HGSOCs) bearing *BRCA1/2* mutations [4,5,6]. It is noteworthy that PARP inhibitors have also contributed to a significant improvement of survival rates in patients with wild-type ovarian cancer, yet still with less efficacy than in patients with *BRCA1/2m* ovarian cancer [4,5,6].

Approximately 50% of HGSOCs are shown in The Cancer Genome Atlas (TCGA) molecular analysis to harbor HRD deficiency, including somatic *BRCA1/2m* (*sBRCA1/2m*) and alterations in other genes essential for the homologous recombination repair pathway such as *ATM*, *ATR*, and *RAD51C/D* [7]. It has been shown that tumor testing is efficient in identifying patients with *BRCA1/2m*, showing high concordance with germline mutation sequencing [8]. Thus, identifying *BRCA1/2* germline and somatic mutations is now essential in routine clinical practice to propose a PARP inhibitor to patients at first relapse, as this is the best predictive biomarker for PARP inhibitor efficacy. With the recent positive results of the SOLO1 phase III trial, it has become increasingly urgent to have *BRCA1/2m* rapid testing results for all patients with newly diagnosed HGSOCs in order to select patient for PARP maintenance after platinum-based first-line therapy [9].

Approximately 95% of HGSOCs have a clonal somatic TP53 mutation (*TP53m*) [7]. Combined *tBRCA/TP53m* testing may provide the advantage of rapid results in comparison to *gBRCA1/2* mutation testing via oncogenetic counseling. This approach may also be useful to interpret *sBRCAm* in low-cellularity samples and provide indirect evidence of the second hit event at the tumor level, such as the loss of heterozygosity (LOH). Evidence suggests that LOH may be a useful biomarker to predict primary resistance to DNA-damaging agents in *BRCA1/2m* carriers [10]. Recent reports of LOH analysis in the *BRCA1/2* locus confirmed a proportion of loss of the wild-type (WT) allele in ovarian tumors ranging from 75% to 93% [10,11,12].

At Gustave Roussy (Villejuif, France), every patient with a new diagnosis of HGSOC (and fallopian or peritoneal carcinoma) is referred to a genetic consultation for counseling and germline testing, along with *tBRCA1/2* mutation testing using next-generation sequencing (NGS) via a dedicated academic platform. This study compares the performance of combined *tBRCA/TP53m* testing to germline testing of ovarian cancer patients seen at Gustave Roussy.

## 2. Results

From 1 January 2016 to 1 May 2018, 237 patients with HGSOCs underwent *tBRCA/TP53m* testing by NGS (Figure 1). These patients were also assigned to a dedicated genetic consultation for *gBRCA1/2* testing, but, for some of them, germline testing results were pending.

Baseline characteristics of the cohort are summarized in Table 1. The median age of patients was 62 years old (IQR 56–68). Most tumors were HGSOCs with stage III or IV at diagnosis.

*gBRCA1/2m* status was available for 189 (79.7%) patients, while it was either still pending or not available for 48 (20.3%) patients (Figure 1). Of these 189 with available status, 27 (14.3%) *gBRCA1m* and 12 (6.3%) *gBRCA2m* were identified.

*tBRCA1/2* testing was performed on the 237 cases with a median testing turn-around time of 37 days (IQR 26.0–52.0 days). Analysis was non-contributive for 41 (17.3%) samples. Reasons for non-contributive samples were mainly poor tumor cellularity and sample quality (Wilcoxon rank-sum test, *p* < 0.001). Heterogeneity of tumoral cellularity was observed among all samples (mean tumoral cellularity of 62%; 3–100%). There was no difference between non-contributive or contributive tumor samples regarding proportions of tumor samples from untreated versus post-neoadjuvant chemotherapy samples (χ2 test, *p* = 0.69). Furthermore, no significant differences were observed between samples collected from biopsies or debulking surgical samples (χ2 test, *p* = 0.37).

Among the 196 contributive samples, 43 (22.1%) *BRCA1/2m* were identified using tumor-based sequencing (Table 2).

All 39 (N = 39/39) known germline mutations were identifiable with tumor-based testing, including one large-scale *BRCA1* rearrangement.

With tumor-based testing, four additional *BRCA1/2* mutations were also identified, and 124 were cases that were confirmed as *BRCA1/2* germline WT.

Among these 43 samples with tumor-based *BRCA1/2* mutations identified, the analysis of LOH was conclusive for 39 samples (Figure 1). LOH was identified in 35 (90%) of them: 24 out of 29 (83%) and 11 out of 14 (79%) for the *BRCA1* and *BRCA2* mutations, respectively (Table 2).

A number of variants of unknown signification (VUS) were also identified: 6 tumoral *BRCA1* VUS and 18 tumoral *BRCA2* VUS (Appendix A).

*TP53m* status was identified using NGS for 184 samples (77.6%) (Figure 2 and Appendix A). We found that 169 of samples tested (91.8%) carried a *TP53m*. The different *TP53m* were 102 missense (60%), 34 frameshift (20%), 20 nonsense (12%), and 13 splicing (8%) (Table 3).

*TP53m* AF was a good control to confirm tumor DNA, with a median *TP53* AF mutation of 0.52 (range 0.01–0.93), confirming a huge variability in tumor cellularity among samples.

Among germline *BRCA* mutation cases, AF ratio of *BRCA1/2m*:*TP53* mutation was superior to 1 in 87% of cases (N = 20/23 of cases), confirming germline origin and suggesting LOH (median ratio 1.3, IQR 1.1–1.9).

The AF *BRCA1/2m/TP53m* ratio was lower among identified somatic *BRCA1/2m* tumor samples (median AF *BRCA1/2m/TP53m* ratio = 1.0, IQR 0.9–1.4) but always >0.7, suggesting that acquired *BRCA1/2* mutation is clonal and associated with LOH.

For three *gBRCA1/2* wild-type samples with <10% cellularity and very low deletion of *BRCA1/2m* AF (0.04, 0.04, and 0.08), *TP53* AF was also <0.05, thus validating somatic *BRCA1/2* mutation in these cases.

## 3. Discussion

It now seems clearly established that for every patient with newly diagnosed HGSOC, the mutational status of *BRCA1/2* should be determined at diagnosis. In the context of the recently published results of phase III SOLO1, it also now seems mandatory to obtain the *BRCA1/2* status as soon as possible to propose a PARP inhibitor in maintenance treatment for patients with stage III–IV in complete response after initial debulking surgery followed by first-line platinum-based chemotherapy [13]. Furthermore, FDA and EMA have recently given their favorable approval to PARP inhibitors regardless of *BRCA1/2* status. However, the information remains crucial as the magnitude of the benefit from maintenance PARP inhibitors in first-line treatment varies greatly for *BRCA1/2m* versus *BRCA1/2* WT patients. Germline *BRCA1/2* testing can be more complex to organize as access to genetic counseling is required. Starting the analysis by tumoral *BRCA1/2* screening can facilitate access to results since tumoral samples can be directly analyzed without any prior genetic counseling.

The first advantage of tumor-based *BRCA1/2* testing is that the testing turn-around time is significantly reduced, with a median of 37 days observed, making it suitable for clinical use in practice.

Secondly, tumor-based *BRCA1/2* testing is as sensitive as blood-based testing for germline variants as we could identify all the known *BRCA1/2* germline mutations, including a large rearrangement. Additional *sBRCA1/2m* were also identified, providing additional information about factors such as LOH presence.

The results were consistent with previously published studies regarding the efficiency of *tBRCA1/2* testing in clinical practice. As an example, the PAOLA-1 study showed rates of non-contributive samples of around 15% to 18% using academic platforms [14]. The number of non-contributiveness samples seems to be high, which could be related more to older material than current practice. It is also important to reject any low-quality sample to avoid the risk of a false negative in the result.

Another noteworthy point is that assessment of *TP53* mutational status, along with *BRCA1/2*, seems to be a good quality control for validating the tumor cellularity of samples, and it is essential for good interpretation of the results. Moreover, with combined tumor-based *BRCA1/2* and *TP53* testing, we could also validate the presence of somatic *BRCA1/2* mutations in samples with a low cellularity.

A number of studies confirmed that PARP inhibitors are also effective in some patients with BRCA*wt* HGSOC [6,14,15,16]. Whether mutations in HRD pathway genes, such as *RAD51C/D* or *PALB2* [17], or the methylation in *BRCA1* or *RAD51C* promoters can identify HRD tumors that would benefit from PARP inhibitors is worthy of investigation [18,19]. The position of *TP53* mutational status and its allelic fraction could also be an important marker to correctly interpret those results.

Finally, tumor-based testing at progression could be particularly valuable for uncovering acquired resistance mechanisms to PARP inhibitors, such as secondary reversion mutations in *BRCA1/2* or *RAD51* genes resulting in restoration of homologous recombination function [20].

## 4. Materials and Methods

The authors reported all consecutive cases of HGSOCs with tumor-based *BRCA1/2m* testing that were treated at Gustave Roussy (Villejuif, France) from 1 January 2016 to 1 May 2018. All patients with HGSOC were referred to a dedicated genetic consultation to determine germline *BRCA1/2* (*gBRCA1/2*) mutational status. Tumor-based *BRCA1/2* testing was prospectively performed using NGS panels (SureSeq Ovarian Cancer Panel (Oxford Gene Technology—7 genes)) and a customized SureSelect XT HS homemade panel (Agilent Technology—12 genes).

Tumor samples used for *BRCA1/2* testing were either samples available at diagnosis or at relapse and were collected either from biopsies at diagnostic laparoscopies or samples from upfront or interval debulking surgery. Pre-treatment samples were preferred. The AF ratio for *BRCA1/2* and *TP53* mutations was calculated to estimate the proportion of cells carrying the *BRCA1/2* mutation and to detect the presence of LOH. A tumor sample was said to have LOH if the *BRCA1/2* variant allelic fraction was greater than 60%. For those samples whose *BRCA1/2* allelic frequency was below 50%, the authors concluded that there was an LOH only if the *BRCA1/2* allelic fraction was similar to that of *TP53* mutation.

Univariate analysis, Wilcoxon rank-sum test, Fisher’s exact test, and the χ2 test were used for comparisons of patient characteristics and mutational status when appropriate. A two-sided *p*-value <0.05 was considered statistically significant for all analyses.

All statistical analyses were performed using R (version R 3.2.2, Copyright© 2004–2013).

## 5. Conclusions

In conclusion, combined tumor-based *BRCA1/2* and *TP53* testing is sensitive for the identification of both somatic and germline *BRCA1/2* mutations and feasible in routine practice with an acceptable turn-around time. Additionally, the *TP53* AF provides useful information regarding sample tumor cellularity and LOH that can help better identify *sBRCA1/2m* in low-cellularity samples.

## Figures and Tables

**Figure 1 ijms-24-11570-f001:**
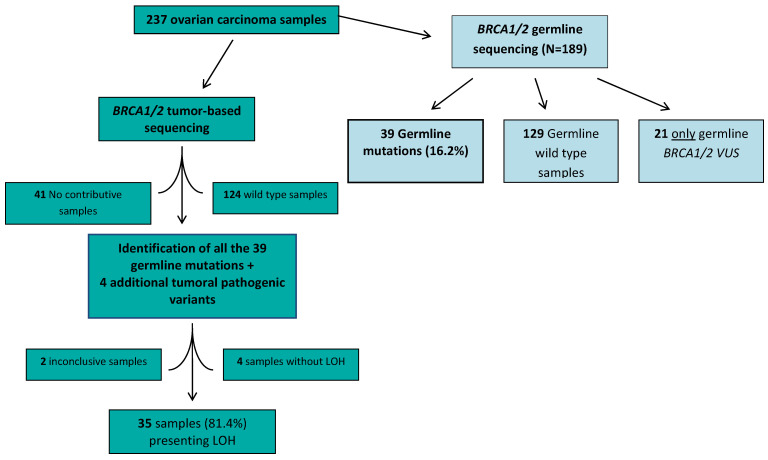
Flow chart of tumor-based *BRCA1/2* and germline testing.

**Figure 2 ijms-24-11570-f002:**
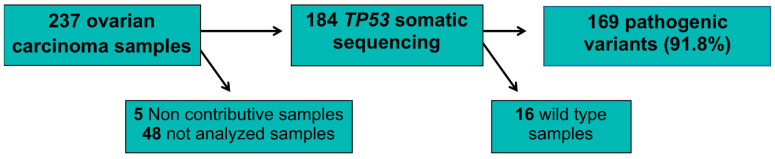
*TP53* tumor samples testing.

**Table 1 ijms-24-11570-t001:** Baseline characteristics of patients who underwent *BRCA1/2* tumor-based testing.

	n = 237 Patients
Median age at diagnosis (IQR) *	62.0 (56.0–68.0)
Histological type, n (%)	
- High-grade serous carcinoma	205 (86.5)
- Low-grade serous carcinoma	4 (1.7)
- Clear-cell carcinoma	2 (0.8)
- Carcinosarcoma	4 (1.7)
- High-grade endometrioid carcinoma	15 (6.3)
- Undifferentiated carcinoma	7 (3.0)
FIGO stage, n (%)	
- I	8 (3.3)
- II	10 (4.2)
- III	126 (53.2)
- IV	45 (19.0)
- NA	48 (20.3)
Type of samples, n (%)	
- Biopsy	93 (39.2)
- Surgical samples	127 (53.6)
- NA	17 (7.2)
Sample collection, n (%)	
- Primitive	141 (59.4%)
- Primitive post-neoadjuvant treatment	62 (26.2%)
- Relapse	1 (0.4%)
- Relapse post chemotherapy	27 (11.4%)
- NA	6 (2.5%)

Interquartile range: IQR. * Clinical data were unavailable for 46/237 patients.

**Table 2 ijms-24-11570-t002:** *BRCA1/2* mutations identified in tumor testing and their respective LOH analysis.

Gene	Variant	Protein	Functional Domain	Variant Type	Germline	Allelic Frequency	LOH	*TP53* Associated Mutation	Allelic Frequency
*BRCA1*	c.134+3A>C	-	-	Splicing	No	0.43	Yes	c.375+1G>T	0.45
*BRCA1*	c.1121del	p.Thr374fs	-	Frameshift	Yes	0.7	Yes	c.394A>G; p.Lys132Glu	0.63
*BRCA1*	c.212+3A>G	-	Ring finger	Splicing	Yes	0.85	Yes	c.673-1G>C	0.55
*BRCA1*	c.1674del	p.Gly559Valfs*13	-	Nonsense	Yes	0.67	Yes	c.742C>T; p.Arg248Trp	0.57
*BRCA1*	c.68_69del	p.Glu23fs	Ring finger + NES1	Frameshift	Yes	0.46	Inconclusive	No	WT
*BRCA1*	c.190T>C	p.Cys64Arg	Ring finger	Missense	No	0.19	Yes	c.578A>T; p.His193Leu	0.24
*BRCA1*	c.5266dup	p.Gln1756_Asp1757fs	Linker	Frameshift	Yes	0.77	Yes	c.351del; p.Gly117fs	0.29
*BRCA1*	c.5468-2A>G	-	BRCT2/AD2	Splicing	Yes	0.50	No	c.403T>C; p.Cys135Arg	0.18
*BRCA1*	c.5266dup	p.Gln1756_Asp1757fs	Linker	Frameshift	Yes	0.74	Yes	c.840A>C; p.Arg280Ser	0.62
*BRCA1*	c.514C>T	p.Gln172*	-	Nonsense	No	0.04	Yes	c.375+5del	0.05
*BRCA1*	c.81-1G>C	-	Ring finger + NES1	Splicing	No	0.14	Yes	c.518T>C; p.Val173Ala	0.09
*BRCA2*	c.2612C>A	p.Ser871*	-	Nonsense	Yes	0.89	Yes	NR	NR
*BRCA1*	c.815_824dup	p.Gly275_Thr276fs	-	Frameshift	No	0.69	Yes	c.743G>A; p.Arg248Gln	0.61
*BRCA2*	c.6533_6542del	p.His2178Glnfs*10	-	Deletion	No	0.24	Yes	c.1024C>T; p.Arg342Ter*	0.29
*BRCA1*	c.4183C>T	p.Gln1395*	Coil coiled/AD1	Nonsense	Yes	0.62	Yes	c.527G>T; p.Cys176Phe	0.44
*BRCA1*	c.1789G>T	p.Glu597*	-	Nonsense	No	0.16	Yes	c.742C>T, p.Arg248Trp	0.15
*BRCA1*	c.3001G>T	p.Glu1001*	-	Nonsense	Yes	0.68	Yes	c.395A>G; p.Lys132Arg	0.27
*BRCA1*	c.5503C>T	p.Arg1835Stop	BRCT2/AD2	Nonsense	Yes	0.12	Yes	c.395A>G; p.Lys132Arg	0.20
*BRCA1*	c.523A>T	p.Lys175*	-	Nonsense	No	0.39	No	c.614A>G; p.Tyr205Cys	0.54
*BRCA2*	c.7952del	p.Arg2651fs	Helical domain	Frameshift	No	0.47	Yes	c.681_682insT; p.Ser227_Asp228fs	0.42
*BRCA1*	c.4065_4068del	p.Asn1355fs	AD1	Frameshift	Yes	0.81	Yes	c.824G>A; p.Cys275Tyr	0.62
*BRCA1*	c.2389G>T	p.Glu797*	-	Nonsense	No	0.41	Yes	c.586C>T; p.Arg196*	0.26
*BRCA2*	c.6591_6592del	p.Thr2197fs	-	Frameshift	Yes	0.87	Yes	NR	NR
*BRCA2*	c.2612C>A	p.Ser871*	-	Nonsense	No	0.48	No	c.394A>G; p.Lys132Glu	0.15
*BRCA2*	c.8487+1G>A	-	-	Splicing	No	1	Yes	No	WT
*BRCA2*	c.3785C>G	p.Ser1262*	-	Nonsense	Yes	0.28	Yes	c.1025G>A; p.Arg342Gln	0.35
*BRCA2*	c.2612C>A	p.Ser871*	-	Nonsense	Yes	0.69	Yes	c.524G>A; p.Arg175His	0.42
*BRCA1*	c.4658del	p.Leu1553fs	AD1	Frameshift	Inconclusive	0.87	Yes	c.376-2A>G	0.70
*BRCA1*	c.3257T>G	p.Leu1086*	-	Nonsense	Yes	0.55	No	c.824G>A; p.Cys275Tyr	0.22
*BRCA1*	c.1063A>T	p.Lys355*	-	Nonsense	No	0.61	Yes	c.396G>T; p.Lys132Asn	0.64
*BRCA1*	c.3008_3009del	p.Phe1003fs	-	Frameshift	Yes	0.80	Yes	c.1010G>T; p.Arg337Leu	0.52
*BRCA1*	c.2679_2682del	p.Lys893fs	-	Frameshift	Yes	0.63	Yes	c.818 G>T; p.Arg273Leu	0.48
*BRCA2*	c.2492del	p.Val831fs	-	Frameshift	No	0.21	Yes	c.818G>T; p.Arg273Leu	0.23
*BRCA1*	c.4072G>T	p.Glu1358*	AD1	Nonsense	No	0.60	Yes	c.403T>G; p.Cys135Gly	0.62
*BRCA1*	c.3995del	p.Gly1332fs	AD1	Frameshift	No	0.28	Yes	c.783-1_784delinsCTT; p.?	0.20
*BRCA1*	c.4868C>G	p.Ala1623Gly	AD2	Missense	No	0.089	Yes	c.644G>T, p.Ser215Ile	0.0587
*BRCA1*	c.4658del	p.Leu1553fs	AD1	Frameshift	Yes	0.87	Yes	c.376-2A>G	0.70
*BRCA2*	c.5345_5346del	p.Gln1782fs	-	Frameshift	Yes	0.66	Yes	c.307_308insGAAAACCT; p.Tyr103_Gln104fs	0.32
*BRCA2*	c.3233_3234insT	p.Val1078_Ser1079fs	-	Frameshift	Yes	0.76	Yes	c.262del; p.Ala88fs	0.68
*BRCA2*	c.5682C>G	p.Tyr1894*	-	Nonsense	Yes	0.91	Yes	No	-
*BRCA2*	c.1834G>T	p.Glu612*	-	Nonsense	No	0.04	Yes	c.388C>T; p.Leu130Phe	0.0294
*BRCA2*	c.413_417del		-	Frameshift	No	0.02	Inconclusive	No	-
*BRCA1*	Deletion exon 21 to 24	p.?	-	Frameshift	Yes	0.8 *	Yes	c.151del; p.Glu51fs	0.52

* ratio at 0.2 in the deletion/estimation of the allelic frequency.

**Table 3 ijms-24-11570-t003:** Description of *TP53* variant type.

*TP53* Variant Type	Frequency
Missense	102 (60%)
Frameshift	34 (20%)
Nonsense	20 (12%)
Splicing	13 (8%)
Total	169

## Data Availability

The datasets generated during and/or analyzed during the current study are available from the corresponding author upon reasonable request.

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
