# Peer review of "Combined Tumor-Based BRCA1/2 and TP53 Mutation Testing in Ovarian Cancer"

_ijms, 2023, doi:10.3390/ijms241411570_

Round 1
Reviewer 1 Report
Comments and Suggestions for Authors
Manuscript entitled " Combined tumor-based BRCA1/2 and TP53 mutation testing in ovarian cancer" written by Borcoman et al aims to provide the information regarding combined tumor-based BRCA1/2 and TP53 testing to identify somatic/germline BRCA1/2 mutation. This manuscript is well written and authors provide the information very well. I have following concerns:
1) Could authors please distribute the mutations information in different classification schemes like structural, functional and hotspot classification ?
2)Please include the information of institute "Gustave Roussy" including state, country because author mentioned samples were collected from here.
3)Could authors please include any information of patients regarding the Stage of cancer or number of patients under treatment during this study? or any history of carcinoma in these patients ?
4) Did authors collect the data to measure overall survival rate from the time of diagnosis ?
5)Line 178-179: How authors calculated ration of allelic fractions (AF) ? Did they use any specific reference, please include.
Author Response
Could authors please distribute the mutations information in different classification schemes like structural, functional and hotspot classification ?
ïƒ All variants cited in the text are listed in tables inserted in the text and in supplementary material. But if you deem it really necessary, we can provide the lolliplots.
2)Please include the information of institute "Gustave Roussy" including state, country because author mentioned samples were collected from here.
ïƒ We included the information requested. (Villejuif, France)
3)Could authors please include any information of patients regarding the Stage of cancer or number of patients under treatment during this study? or any history of carcinoma in these patients ?
ïƒ All patients included in this study had ovarian carcinoma, Patients characteristics, including stage and hystological subtypes are specified in Table 1.
4) Did authors collect the data to measure overall survival rate from the time of diagnosis?
ïƒ We added the survival curves in the supplementary data file. PFSm and OSm were 2.3y months and 14.4y respectively.
5)Line 178-179: How authors calculated ration of allelic fractions (AF) ? Did they use any specific reference, please include.
ïƒ There is no specific literature reference for the computation of allelic fraction. The allelic fraction is assessed by dividing the number of reads with an alteration divided by the total number of read on the same locus.
Reviewer 2 Report
Comments and Suggestions for Authors
I read your paper with much interest and I think it is well written with a clear message.
Some questions/remarks:
- line 81/82: how is it possible that with a retrospective study with inclusion that ended almost 5 years ago that more than 40% of the germline results are still missing?
- line 104: regarding the VUS you found I would recommend to add if they are class 3 or 4
- line 140/141: you state that with germline BRCA testing the TAT would be too long to be suitable in a clinical context. This is a general statement that is not true as such, as a TAT of 1 month is perfectly feasible and soon enough before the PARPi treatment starts. Please elaborate in details on this claim as this is a core conclusion of your research.
Author Response
REVIEWER2
The article discusses the use of combined tumor-based BRCA1/2 and TP53 testing in identifying somatic and germline BRCA1/2 mutations (BRCAm) in high grade serous ovarian cancers (HGSOC). The testing allows for rapid results and indirect evidence of the "second hit" event, which is loss of heterozygosity (LOH). The article reports that among 237 HGSOC patients tested, 43 deleterious tumor BRCA1/2 mutations (tBRCAm) were identified, with 23 being germline and 20 being somatic. TP53 allelic fractions were calculated to estimate the proportion of cells carrying BRCAm and to infer LOH. The study concludes that combined tumor-based BRCA1/2 and TP53 testing is sensitive and useful in identifying BRCAm and indirect evidence of LOH.
Major issues:
- The paper only focuses on the mutational status of BRCA1/2 and TP53. It will be better if you can expand your analysis to include the correlation of those mutations and changes of gene expression programs.
The screening for TP53 is not proposed in many screenings test for BRCA1/2 since there is no direct therapeutic impact. The correlation on the gene expression is not relevant for TP53 because it evolves in both directions depending on the mutation. For BRCA1/2, there may be a total loss, but not systematic with certain mutations. Therefore, the expression is not a usable information. Besides, we do not have access to RNA samples to explore.
- It will be better if you can explore the biological functional relationship between LOH of those genes and transcriptomic profiles of those samples.
We thank you for this perspective however the quality of FFPE samples and the RNA is not enough available to explore this hypothesis.
- Validating the results using TCGA or other datasets with appropriate number of prognostic information.
We included the TCGA data in the discussion section.
Reviewer 3 Report
Comments and Suggestions for Authors
The article discusses the use of combined tumor-based BRCA1/2 and TP53 testing in identifying somatic and germline BRCA1/2 mutations (BRCAm) in high grade serous ovarian cancers (HGSOC). The testing allows for rapid results and indirect evidence of the "second hit" event, which is loss of heterozygosity (LOH). The article reports that among 237 HGSOC patients tested, 43 deleterious tumor BRCA1/2 mutations (tBRCAm) were identified, with 23 being germline and 20 being somatic. TP53 allelic fractions were calculated to estimate the proportion of cells carrying BRCAm and to infer LOH. The study concludes that combined tumor-based BRCA1/2 and TP53 testing is sensitive and useful in identifying BRCAm and indirect evidence of LOH.
Major issues:
1. The paper only focuses on the mutational status of BRCA1/2 and TP53. It will be better if you can expand your analysis to include the correlation of those mutatoins and changes of gene expression programs.
2. It will be better if you can explore the biological functional relationship between LOH of those genes and transcriptomic profiles of those samples.
3. Validating the results using TCGA or other datasets with appropriate number of prognostic information.
Author Response
REVIEWER 3 –
I read your paper with much interest and I think it is well written with a clear message.
Some questions/remarks:
- line 81/82: how is it possible that with a retrospective study with inclusion that ended almost 5 years ago that more than 40% of the germline results are still missing?
- The germline status was not systematically performed as the screening was first in tumoral. We up-dated our database. So far, ~80% of the patients had access to germline sequencing. This infromation has been updated in the text.
- gBRCA1/2m status was available for 189 (79.7%) patients, while either still pending or not available for 48 (20.3%) patients
- Line 104: regarding the VUS you found I would recommend to add if they are class 3 or 4
- They are all class 3. The supplementary table was modified as proposed.
- Line 140/141: you state that with germline BRCA testing the TAT would be too long to be suitable in a clinical context. This is a general statement that is not true as such, as a TAT of 1 month is perfectly feasible and soon enough before the PARPi treatment starts. Please elaborate in details on this claim as this is a core conclusion of your research.
- We agree your comment about the TAT. The main concern on germline BRCA testing is the management of the patient by the genetic counseling which is not easily available. This can add more time to access the BRCA testing.
- We modified the sentence :” With germinal BRCA1/2 testing, the average time to obtain the mutational status might not be suitable for clinical routine in that context.” To “the germline BRCA1/2 testing can be more complex to organize if the access to genetic counseling is required. Starting the analysis by tumoral BRCA1/2 screening can facilitate the access to results since tumoral samples can be directly analyzed.”

Round 2
Reviewer 3 Report
Comments and Suggestions for Authors
I think all issues have been properly resolved.